# Water Inrush Mechanism and Treatment Measures in Huali Highway Banyanzi Tunnel—A Case Study

Yuanzhi He [1], Hanxun Wang [1,*], Jin Zhou [1], Haifeng Su [2], Li Luo [2] and Bin Zhang [1]

1 School of Engineering and Technology, China University of Geosciences (Beijing), Beijing 100083, China
2 China Construction First Group Corporation Limited, Beijing 100071, China
* Correspondence: hanxun_w@cugb.edu.cn; Tel.: +86-13718463771

**Abstract:** In the process of tunnel construction, the water inrush disaster is one of the main engineering geological disasters. In karst strata, different types of water-bearing structures or karst water bodies develop and occur in the soluble rock layer, and tunnel excavation easily forms new drainage channels, resulting in water inrush in the tunnel. Based on the project of the Huali Highway Banyanzi Tunnel, this paper studies the water inrush characteristics, water inrush mechanism, and treatment measures of the karst tunnel. According to the basic data, combined with field investigation, data monitoring, geological radar detection, tracer test, and numerical simulation, the characteristics and hydrogeological conditions of the tunnel water surge were investigated and analyzed. In addition, the mechanism of tunnel water surge was further summarized. Moreover, the tunnel water-gushing management measures are optimized and verified based on the tunnel water spraying mechanism.

**Keywords:** drainage tunnel; Karst tunnel; numerical simulation; prevention and control measures; water-gushing mechanism

## 1. Introduction

Tunnel construction is an indispensable part of highway construction in mountainous areas. However, due to the complexity of geological and groundwater conditions, tunnel construction and even operation are prone to a large number of geological disasters such as water-gushing mud, collapse deformation, rock burst, gas, and so on. Water inrush disaster is one of the main geological disasters in the process of tunnel construction. Analyzing the formation mechanism of water inrush correctly is the basis for the prevention and control of tunnel water inrush [1–6].

In terms of tunnel water-gushing mechanisms, in the fields related to underground engineering, the coal mine field first began to study the water-gushing situation through the analysis of geological and hydrogeological conditions. In the 1960s and 1970s, the concept of relative water-barrier thickness was introduced in the Mining Safety Regulations of the Hungarian National Mining Technical Appraisal Committee. From the 1970s to 1990s, with the deepening understanding of karst water inrush and the continuous development of related technologies, many countries began to introduce energy methods, system theory, neural networks, and other methods to study the mechanism of karst water inrush [7–10]. In order to clarify the mechanism of tunnel water inrush, various methods, including geological survey, geochemical analysis, drilling, geophysical exploration, and so on, are often combined to investigate the geological and hydrogeological conditions of the tunnel site and then combined with the characteristics of tunnel water inrush analysis [11]. Many scholars used case summary, typical case analysis, laboratory tests, numerical simulation, and other methods to analyze, summarize, and explain the mechanism of tunnel water inrush, including the study of the relationship between water sources and channels [12], the engineering situation of coal seam floor water inrush similar to tunnel water inrush [13], and the water inrush mode of collapse column [14]. Karst is one of the main disaster sources

for tunnel water inrush. A large number of scholars have conducted in-depth research on the problem of tunnel water inrush caused by karst. Based on practical data and theoretical analysis, water inrush in karst tunnels requires geological conditions for developing karst, sufficient filling materials, abundant underground water sources, and disturbance caused by engineering excavation [15]. Many scholars have studied the mechanism of water inrush in karst tunnels from the perspective of practice [16–18], such as water inrush and mud inrush in the Dayaoshan tunnel [19] and the water inrush accident in the Jiudingshan karst tunnel on the Chu-Da Highway [20]. Besides the traffic tunnels, the water transmission tunnels are also facing many challenges such as lining and surrounding rock cracking due to the water-rock and water-lining interactions and pore pressures [21–24]. In addition, the water inrush prevention measures of underwater tunnels and pipe jacking tunnels are also concerned and studied by scholars [25,26].

In terms of the prevention and control methods of tunnel water inrush, since the 1950s, attention has been paid to the treatment technology of tunnel water inrush and has been the subject of relevant exploration. In 1975, the "Railway Engineering Technical Code" clearly put forward that "the comprehensive control measures should be taken to give priority to the discharge and combine the blocking and discharge". In 1986, the Code for Design of Railway Tunnels made it clear that "prevention, interception, discharge and blocking are combined, and measures are taken according to local conditions, and comprehensive treatment" is the principle of water inrush management in tunnels, which reflects the further improvement of awareness of water resource protection. At present, common karst tunnels take the following measures to deal with water inrush: Culverts and drainage tunnel drainage, sewer drainage scheme, lining and grouting water plugging, and blind drainage, moreover also formed a complete set of karst forecast measures, in view of the small cave sealed, backfill, strengthening the lining, or the lintel, measures such as strengthening the tunnel for large crossing scheme of karst groundwater, the top reinforcement scheme, and the corresponding construction methods, Anti-seepage measures of lining [27,28]. However, there are few studies on the optimization of tunnel water-gushing prevention measures. Taking the Banyanzi Tunnel of the Huali Highway as an example, this paper studies the changes in water pressure and tunnel structure deformation after optimizing the water-gushing prevention measures. It is verified that the optimized water-gushing prevention measures can effectively reduce the water pressure of the tunnel, and the tunnel structure deformation is kept within the safe range. It provides a reference for the treatment of tunnel water gushing in the karst area.

## 2. Materials and Methods

The analysis of tunnel water inrush mechanism and prevention and control measures proposed in this paper mainly includes two parts: (1) analysis of tunnel water inrush mechanism and (2) prevention and control measures of tunnel water inrush.

### 2.1. Analysis of Tunnel Water Inrush Mechanism

In order to find out the mechanism of water inrush in the tunnel, the main factors affecting water inrush in the tunnel must be analyzed. The analysis process is as follows:

(1)    Analysis of karst development

Geological radar was used to detect karst development around the tunnel. The development of karst in the mountain was studied by tracer tests. Sodium chloride was used as the tracer, and two groups of tracer tests were conducted. In the first group of tracer tests, the tracer was placed at the gully and collected in the tunnel. In the second group of tracer tests, the tracer was put into the tunnel and collected at the spring hole. The concentration of chloride ions in the collected samples was analyzed to study the connectivity of the mountain gully, tunnel water inrush cavity, and spring water points around the mountain.

(2)    Analysis of water inrush characteristics in the tunnel

The water inrush in the tunnel was analyzed by calculating the water inrush at the site and observing the seepage position of the tunnel, the damage to the tunnel structure, and the phenomenon of calcium carbonate crystallization. The rainfall and tunnel water inrush were monitored, and the relationship between rainfall and tunnel water inrush was studied.

*2.2. Prevention and Control Measures of Tunnel Water Inrush*

According to the water inrush mechanism of the tunnel, reasonable water inrush prevention measures are adopted, and a numerical simulation is carried out to analyze the changes in water pressure distribution and displacement distribution of the tunnel and surrounding rock under different working conditions. The main steps are as follows:

(1)    A three-dimensional terrain and tunnel model was constructed, and the location of karst caves in the model was arranged according to the results of geological radar detection. Different water pressure boundaries are set according to different working conditions.

(2)    The section that passes through the karst cave and is perpendicular to the tunnel axis is selected to analyze the calculation results, and different monitoring sites of the tunnel are selected to analyze the changes in water pressure distribution and displacement distribution of the tunnel and surrounding rock under different working conditions, so as to verify the effectiveness of the water inrush prevention measures.

*2.3. Overview of Karst Area*
2.3.1. Engineering Geological Conditions

The Huali Highway Banyanzi Tunnel is located in Yongsheng County, Lijiang City, Yunnan Province, which is located in the middle of Lijiang City, about 103 km away from Lijiang City(Figure 1). The tunnel is a two-way separated long highway tunnel, the length of the left line is 3190 m, the starting point of the left line and most of the tunnel body are located on the circular curve of a radius of 3180 m, the length of the right line is 3220 m, the starting point of the right line and most of the tunnel body are located on the circular curve of a radius of 3000 m, with the difference in mileage, the tunnel spacing varies at the starting point and exit of the tunnel. The spacing between the left and right lines is small, about 42 m; at a distance from the tunnel entrance and exit, the spacing between the left and right lines is about 60 m at most. The tunnel is located at a +2.35% one-way slope. The maximum depth of the tunnel is close to 590 m. The tunnel site belongs to the middle and low mountain carbonate structure denudation (dissolution) geomorphic area. The midline elevation in the tunnel range is about 1660~2290 m, and the natural slope of the mountain is about 45°~65°. The vegetation on the top of the mountain is mostly pine, and sporadic weeds develop on the hillside. The lower part of the mountain is the horse river, and the tunnel entrance is located downstream of the river. The tunnel is located at the junction and transition of the Songpan-Ganzi geosynclastic fold system and the Yangtze quasi-platform, spanning three secondary units, namely the Zhongdian fold belt, the salt rock-Lijiang platform margin fold belt, and the Kang–Yunnan axis. Since the Sinian system, the area has experienced many tectonic changes of different paradigms, and the geological structure is very complex. The fault structure in the route area of the tunnel is very developed, and there are five regional faults. Among them, the F4 Lagude-Zhumadi fault has a great impact on the tunnel: it is a left-lateral thrust fault with a length of about 64 km, cutting Devonian and Triassic strata. The hanging wall is Devonian ($D_2$) Zage Formation dolomite, and the footwall is Triassic ($T_3{}^s$) Shezi Formation sandstone and argillaceous sandstone. The fault strike is 170°, and the fault plane is 80°∠30°. Geophysical exploration results show that the fault fracture zone is about 40 m wide and buried under the overburden of the Quaternary system. The fault does not directly intersect with the tunnel. It is about 580 m away from the tunnel exit, and the intersection angle is nearly 90°. It was an active fault in the middle Pleistocene, and the activity is not obvious after the late Pleistocene. The

F5 left-lateral thrust fault occurs at 160°∠75° on the fault plane and extends in a soothing wave shape. Both sides of the fault plane are Devonian strata. The fault is about 1400 m from the tunnel entrance, and the angle of the intersection is about 63°. It is an inactive fault in Holocene. According to the results of geophysical exploration, the width of the fault fracture zone is about 60 m, and the influence range is about 250 m, which has a great influence on the surrounding rock of this section. Under the action of tectonic stress, structural joints and fractures are of large scale and general ductility, often parallel to or associated with regional faults, with good penetration, relatively stable occurrence, and a flat, smooth, and tight joint plane. According to geological mapping and engineering drilling, the overlying strata in this area are mainly quaternary residuals ($Q_4^{el+dl}$) silty clay and macadam soil, alluvial ($Q_4^{al+pl}$) sand and pebble beds along the river section, and locally there are residual ($Q_4^{el}$) red clay, colluvial ($Q_4^{col}$) macadam soil, and artificial fill soil ($Q_4^{ml}$). The underlying bedrock is mainly sedimentary and metamorphic rocks, which are mainly sandstone, argillaceous sandstone, shale, mudstone, and dolomite from the Cretaceous, Jurassic, Triassic, Devonian, Cambrian, and Lower Proterozoic(Figure 2). Coal lines and oil shale are locally developed. The surrounding rocks near the water-gushing section of the tunnel are middle Devonian limestone and dolomite.

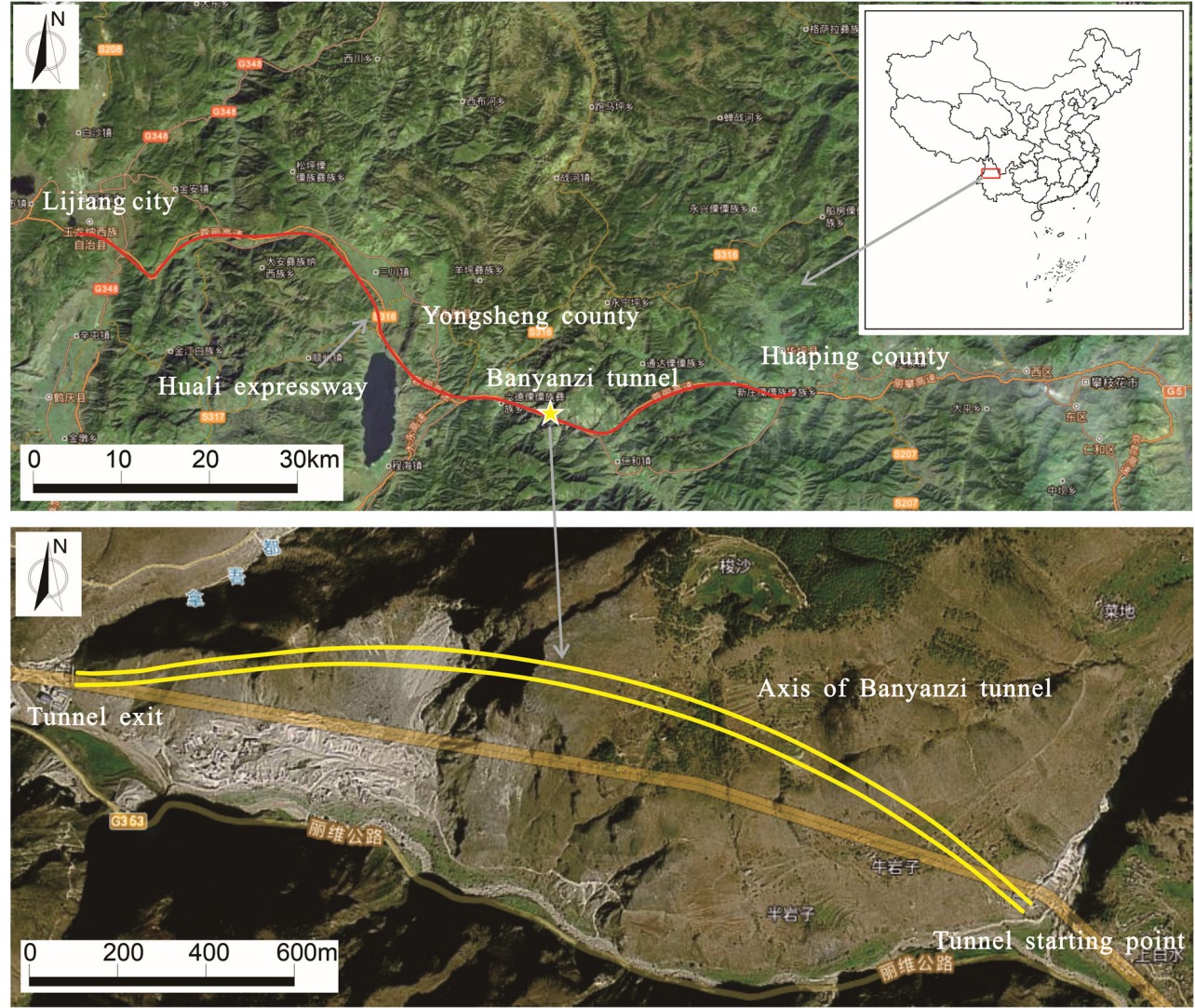

**Figure 1.** Geographical location of the tunnel.

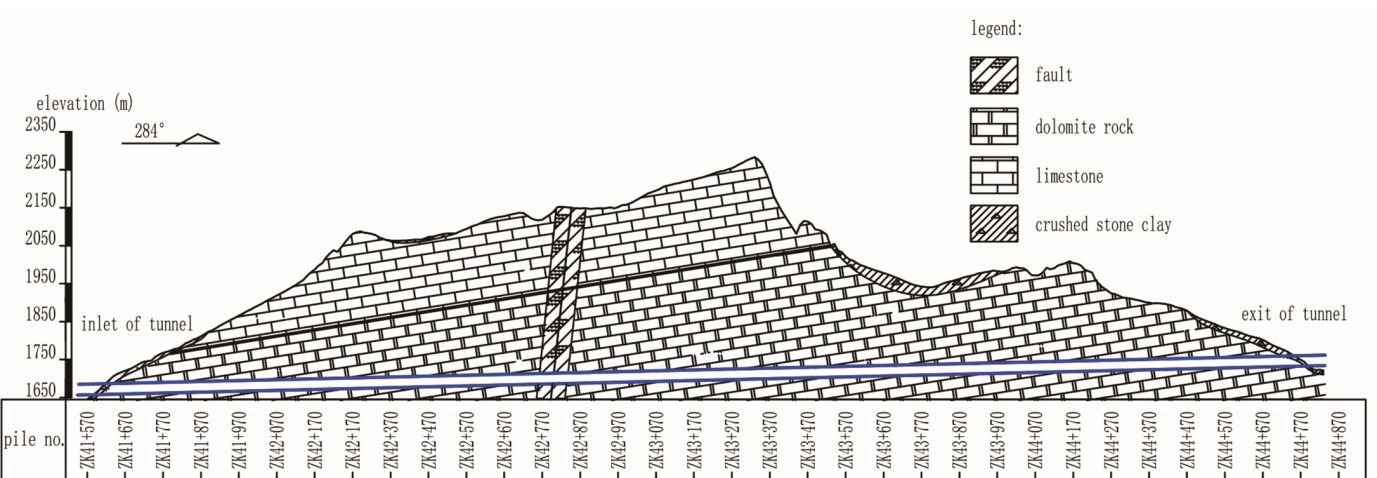

**Figure 2.** Geological profile along the tunnel.

Lijiang City, where the tunnel is located, has a plateau-type southwest monsoon climate with obvious vertical zoning characteristics. According to the local meteorological data, the monthly average minimum temperature appears in January (−0.3 °C), and the monthly average maximum temperature appears in May (28.1 °C). In 2020, the minimum temperature is −5.7 °C, the maximum temperature is 30.3 °C, and the frost-free period is 205 days [29].

2.3.2. Hydrogeological Conditions

The rock body in the study area is generally intact but influenced by geological structures such as faults and folds. Deep tectonic joints are developed in local sections, and fault fragmentation zones, fold cores, and joint development zones provide more favorable conditions for groundwater distribution. According to the analysis of the geological survey, physical exploration results, tracer tests, and existing geological data, the study area is considered an independent hydrogeological unit.

After further analysis of the hydrogeological conditions in the tunnel site area through site investigation, geological radar sounding, and tracer tests, the groundwater in the area is divided into three types: loose pore water, bedrock fracture water, and karst water.

The pore water of the loose accumulation layer is mainly distributed in the alluvial layer of the Quaternary Holocene in the study area, mostly in the form of diving, and the stream water is mutually complementary to the drainage relationship, with shallow burial, close to the recharge source, permeability, water-rich, and other characteristics. Within the study area, it is mostly in the form of dripping and infiltrating output. The topography of the study area is undulating and steep; the thickness of the cover layer that can store this type of pore water is thin; this type of groundwater recharge and discharge is fast; the overall water volume is small; and the impact on the tunnel is small.

Bedrock fracture water is mainly distributed in the area of the carbonate sandwich clastic rock group of water-bearing rocks. This rock group's carbonate-clastic rock type solution is not developed; groundwater is mainly stored in the dense zone of joint fractures; and the degree of water richness is weak.

The karst water is mainly distributed in the shallow karst pipes. The carbonate rocks in the study area are mainly Devonian dolomitic tuffs and dolomites. There are muddy tuffs between this rock group in the tunnel area, which leads to the formation of a relatively independent groundwater system in the area, and the richness of its development is directly controlled by the degree of joint development and karst development. According to the geological mapping and physical exploration test results, it is believed that the shallow karst development of the formation, the development of surface solution grooves, the influence of fault fracture zone, the development of karst fissures in the section of joint

fracture development is more developed and water-bearing, local development of small-scale solution cavities, ranging from 20 cm to 80 cm, no large-scale cavities were found in the study area, surface solution grooves, horizontal and vertical cavities and other karst phenomena are seen in the tunnels, influenced by surface rainfall. It is inferred that the surface water is infiltrated along the karst fissures to the low discharge area, and the horizontal and vertical karst channels developed are the transport channels of groundwater.

The groundwater in the study area is mainly karst water, which has a significant impact on the tunnel, followed by bedrock fracture water and pore water in the loose accumulation layer with a small volume and limited distribution, which has a small impact on the tunnel.

There are no rivers, reservoirs, or other large surface water bodies in the study area, and groundwater is mainly recharged by atmospheric precipitation. The weathering joints and fractures on the surface of the study area are developed, and the rock body is broken, which provides favorable conditions for the recharge and storage of groundwater. The degree of recharge is closely related to the strength and duration of precipitation, geological conditions, the exposed area of aquifers, and the degree of development of shallow fissures.

Since the terrain of the study area is relatively steep, with a large topographic slope, longitudinal gully development, and a low erosion datum, it is favorable to groundwater runoff and discharge. After receiving recharge, part of the groundwater is discharged to the valley or surface in the form of scattered flow or springs in the low-lying terrain, while the other part continues to runoff to the lower part of the slope along the fissure zone and finally discharges to the streams and gullies. Seven springs were found during the survey, mainly exposed on the banks of the river next to the mountain, mainly as descending springs. Among them, one spring flows all year round, and the volume of water varies with the season, ranging from 100 to 1500 m$^3$/h. The other six springs are seasonal, with a large volume of water in the rainy season and dry in the dry season, ranging from 1 to 500 m$^3$/h. Because the amount of groundwater is greatly influenced by rainfall, the local climate has obvious dry and rainy seasons, with nearly 80% of the precipitation concentrated from June to September, making the flow rate of groundwater discharge points vary greatly between the rainy and dry seasons. After heavy rainfall, the volume of water in the catchment rushes, streams, and other locations increases rapidly, and the spring flow reaches its highest peak value.

### 2.3.3. Karst Development
Karst Development around the Tunnel

Geological radar was used to detect karst development around the tunnel. Geological radar is a geophysical prospecting method that uses ultra-high frequency narrow pulse electromagnetic waves to detect medium distribution. The transmitting antenna transmits electromagnetic pulse signals underground. If there is a large void behind the tunnel lining and inside the inverted bottom, or if it is not dense or loose, it will form a strong reflection due to the large physical property difference between the tunnel and the surrounding soil and rock. This reflection is used to detect the back of the lining. The main equipment for geological radar detection is the SIR-4000 geological radar detector. During the detection, the horizontal survey line is added to the abnormal area according to the geophysical anomaly of the longitudinal detection line to detect the horizontal distribution of the abnormal area. It is found that the tunnel body passes through many water-rich karst areas, and there are several karst caves around the tunnel. Within 500 m of the exit end of the right tunnel, the areas 155–195 m, 215–245 m, and 345–390 m away from the exit are water-rich fracture development areas. There are four karst caves within 500 m of the exit end of the right tunnel, which are located at the bottom right of the lining 50 m, at the bottom right of the lining 105 m away from the exit, on the right side of the lining 180 m away from the tunnel, and at the bottom left of the lining 295 m away from the tunnel exit. Within 500 m of the exit end of the left tunnel, the area 340–360 m away from the exit and 425 m away from the exit are water-rich fracture zones. There are two karst caves within 500 m of the

exit end of the left cave, respectively, located at the lower right of the lining 490 m away from the exit, and at the right side of the lining 343 m away from the exit(Figure 3).

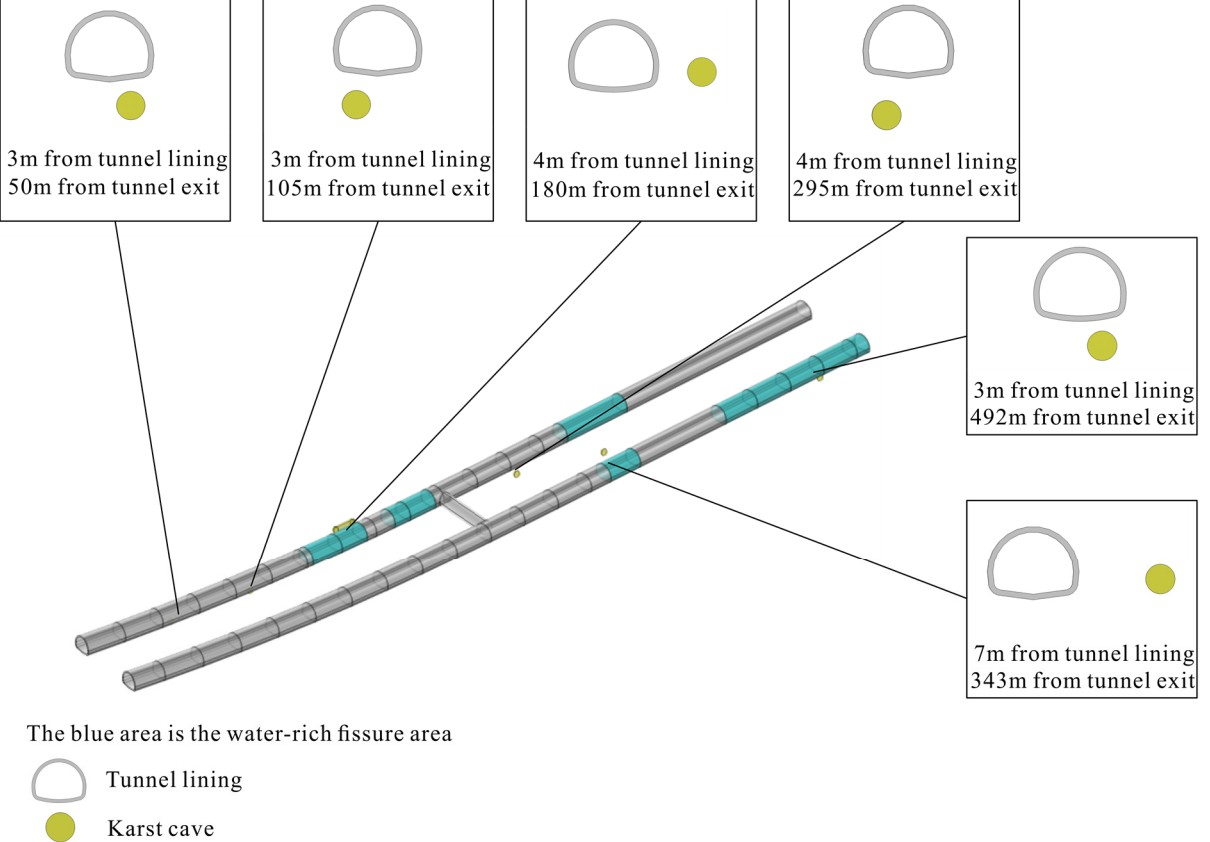

The blue area is the water-rich fissure area

Tunnel lining

Karst cave

**Figure 3.** Schematic diagram of radar survey results.

Development of Karst Inside the Mountain

In order to further study the connectivity of the mountain gully, tunnel water inrush cavity, and spring water points around the mountain, a tracer test was used to study the connectivity. The current tracer test is based on the water flow tracer test, which consists of upstream tracer delivery and downstream tracer monitoring. Sodium chloride was used as the tracer. Two groups of tracer tests were conducted. According to the flow rate of the monitoring sites and the concentration of chloride ions, the detection limit was 10 times the background concentration of chloride ions, and the amount of sodium chloride in the tracer test was 184.6 kg in the first group and 6.6 kg in the second group. In the first group of tracer tests, the tracer was placed in the gully and collected in the tunnel. In the second tracer test, the tracer was placed in the tunnel and collected at the spring (Figure 4).

In the first group of tracer tests, water samples were taken every 10 min at the water inrush of the tunnel, and 10 samples were collected continuously. In the second group of tracer tests, water samples were collected at the spring every 10 min and 10 times continuously. The chloride ions of the collected samples were analyzed, and the test results are shown in Figure 5. The change in ion concentration at the acquisition site after tracer delivery is evident in the figure. The connectivity between the surface water system of the gully above the tunnel and the water inrush point of the tunnel, as well as the connectivity between the water inrush point of the tunnel and the spring hole, is fully proven. Combined with the relationship between rainfall, tunnel water gushing, and spring water gushing, it is basically proven that the tunnel water-gushing point and the spring under the mountain are the drainage points of the gully surface water after entering the mountain.

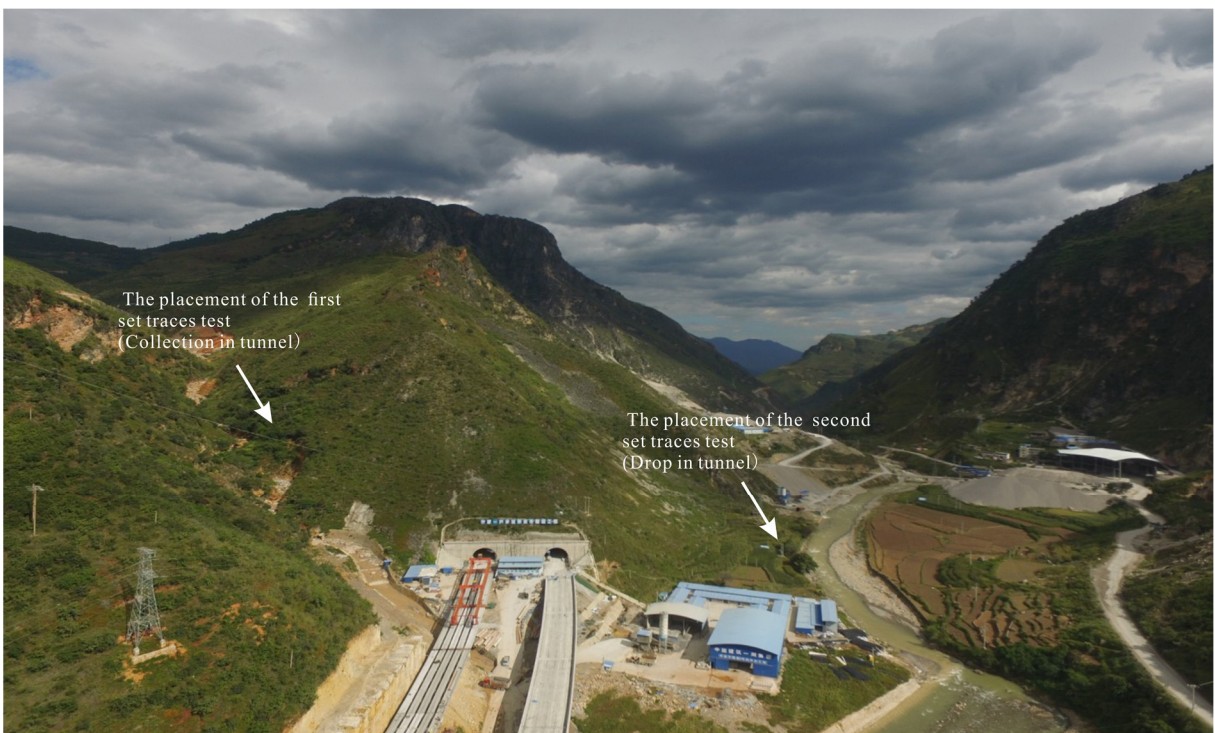

**Figure 4.** Locations of gullies and springs where tracer tests were performed.

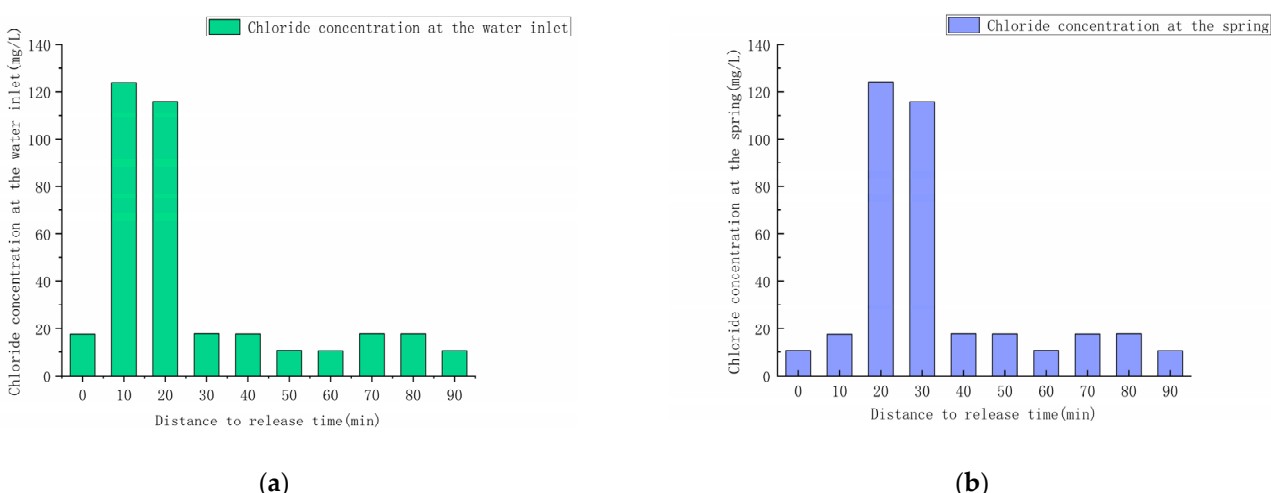

(**a**)                                                                                           (**b**)

**Figure 5.** Tracer test results: (**a**) the first set of results; (**b**) the second set of results.

## 3. Results

### 3.1. Analysis of Water Inrush Mechanism in Tunnel

3.1.1. Characteristics of Water Inrush in Tunnel

Water Inrush in the Tunnel

In July 2017, after continuous rainfall, a water inrush occurred in the tunnel. The water inrush ranges from 75–250 m from the right tunnel to the exit end of the tunnel and 125 m to 200 m from the left tunnel to the exit end of the tunnel, invert, construction joints, and horizontal drainage blind pipe, as well as the vehicle passage and widening belt within the range (Figure 6). The estimated maximum water inflow is about 4666 m$^3$/h. At this time, the tunnel is not yet through, and the face of the exit end of the tunnel has been drilled to about 400 m. It is an anti-slope construction, and the secondary lining and invert have been completed at the water-gushing position.

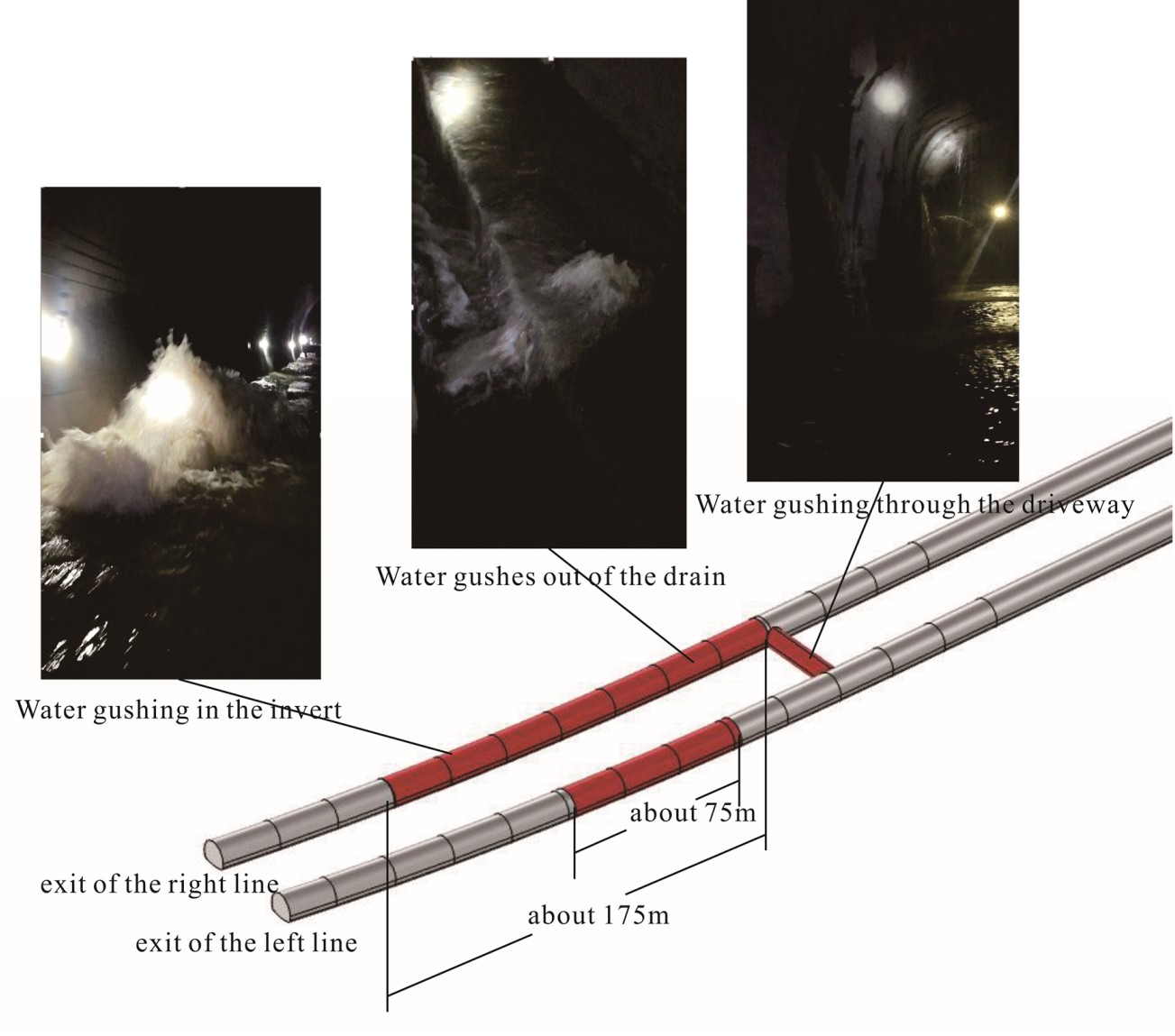

Water gushing through the driveway

Water gushes out of the drain

Water gushing in the invert

exit of the right line

exit of the left line

about 75m

about 175m

**Figure 6.** Water inrush in the tunnel.

In order to reduce the water gushing in the tunnel, a drainage tunnel and drainage and pumping measures are set under the water-gushing section of the tunnel. The contour section area of the drainage tunnel is 12.76 m$^2$. In the second year after the construction of the tunnel, heavy rainfall occurred in the area, with the maximum daily rainfall reaching 87.9 mm. A massive water inrush occurred in the tunnel (Figure 7).

During this process, water seepage occurred in the construction joints of the invert, insufficient water passage capacity of the drainage ditch, calcium carbonate crystals appeared in the annular drainage pipe, and cracks appeared in the secondary lining structure of some water inrush areas, etc. (Figure 8a–d).

Rainfall and Water Inflow Monitoring and Analysis

In order to further understand the characteristics of water inrush in the tunnel, the water inrush in the tunnel was monitored after the water inrush occurred, and the rainfall was monitored at the same time, and the monitoring data of rainfall and water inrush in the tunnel were summarized (Figure 9).

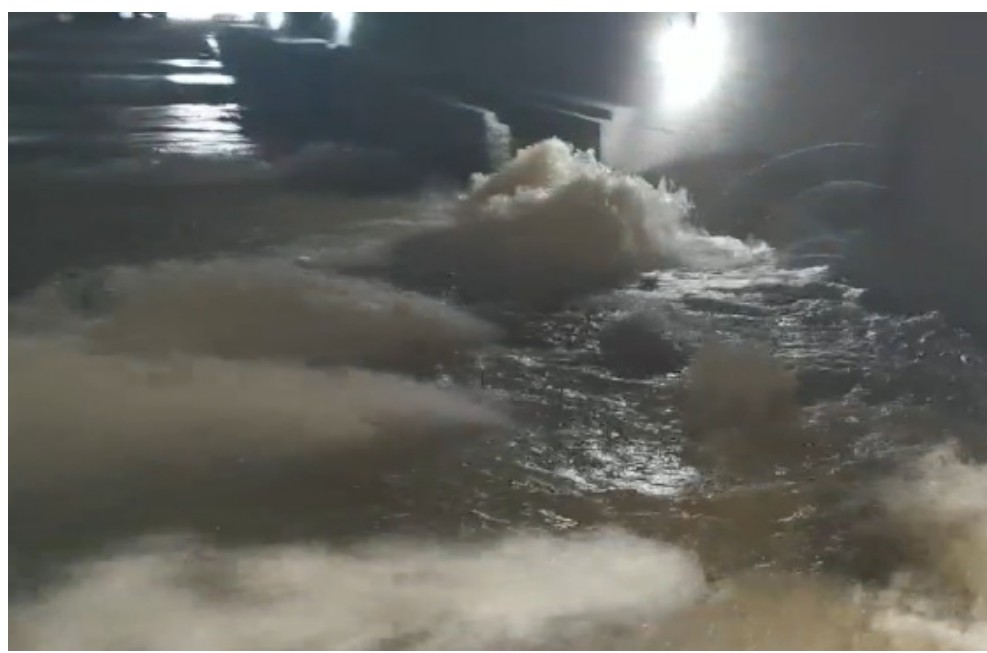

**Figure 7.** Water inrush in the tunnel after initial treatment.

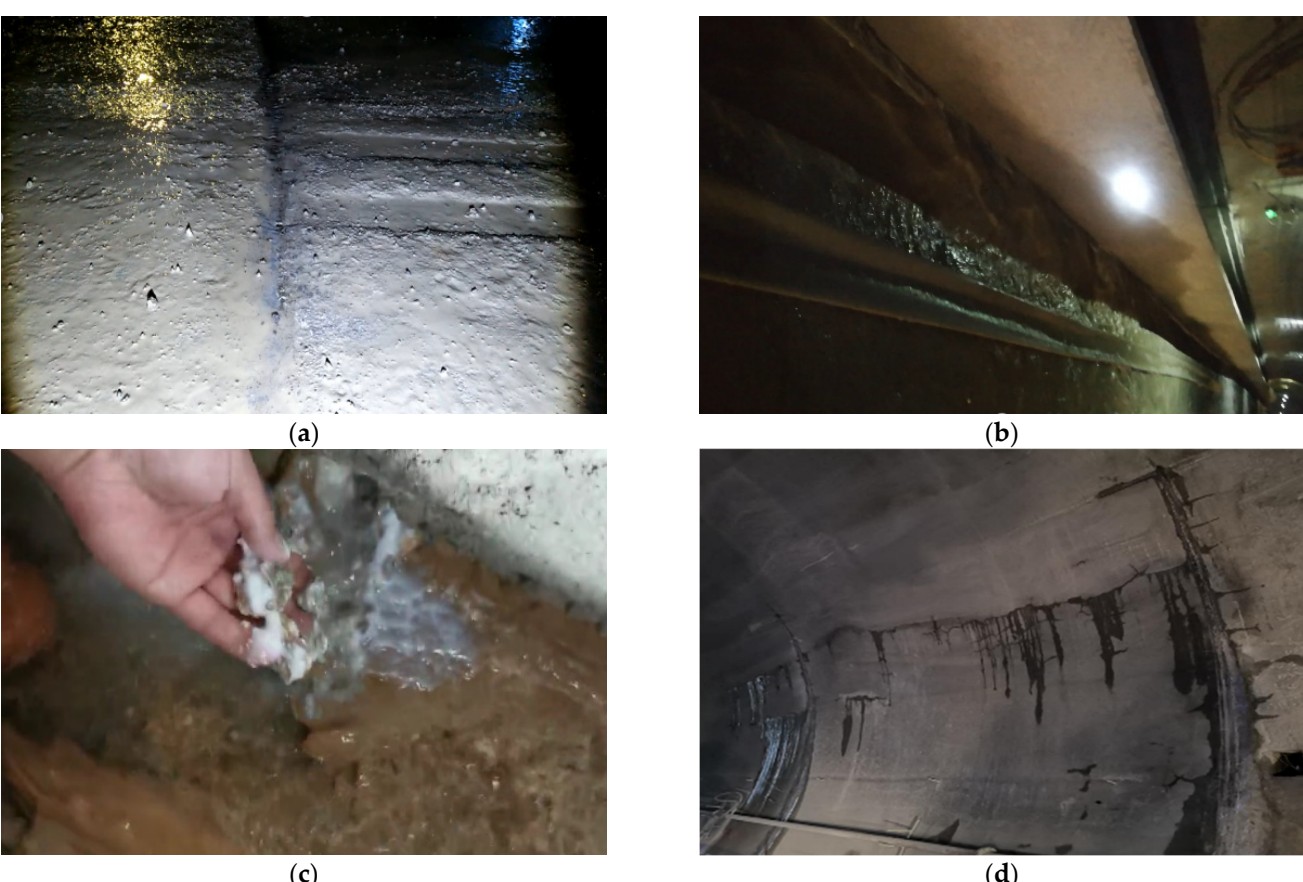

**Figure 8.** (**a**) Water seepage occurs in the inverted arch construction joint; (**b**) inadequate drain capacity; (**c**) calcium carbonate crystals appear in the circumferential drain pipe; (**d**) cracks appear in the secondary lining structure in part of the water-gushing section.

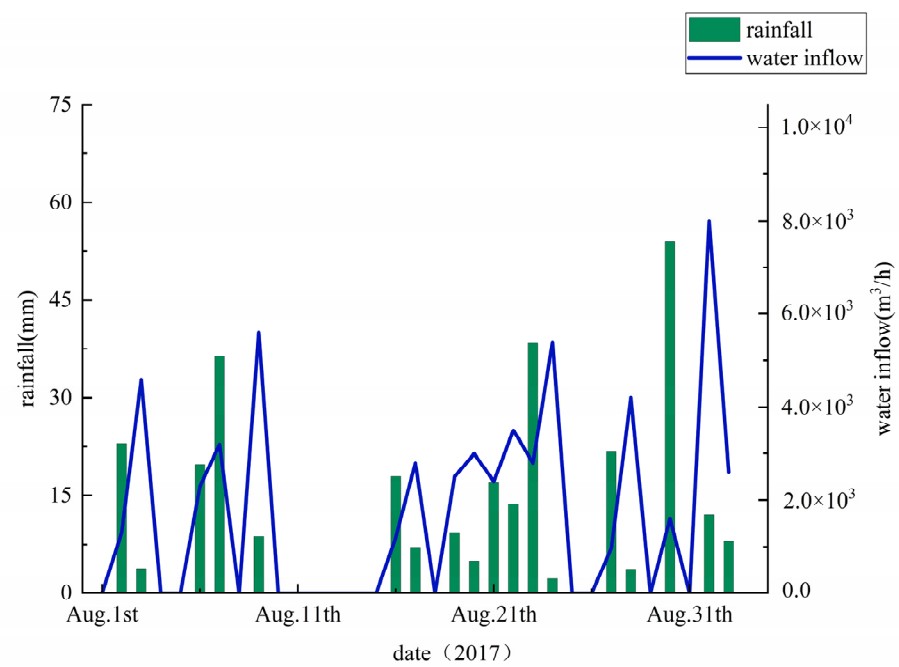

**Figure 9.** Rainfall monitoring—water inflow in the tunnel.

The rainfall measurement location is the Yongsheng Hydrographic station. The monitoring lasted from 2 August 2017 to 2 September 2017 for a total of 32 days, of which rainfall occurred on 18 days, with the maximum daily rainfall of 54 mm, which occurred on August 30. The flood in the tunnel occurred after the rain. Field records showed that water inrush occurred in the tunnel only after rainfall, and a large water volume inrush began to appear in the tunnel about 8 h after heavy rainfall. The maximum daily water inflow was 8000 m$^3$/h, which occurred on September 1, the next day of the maximum rainfall. At the same time, other days with high daily water inrush occurred on the next day of heavy daily rain during the continuous rainfall period. High daily rainfall would not lead to heavy daily water inrush but would lead to large-scale tunnel water inrush in the next 1–2 days even if there was rainfall with a small amount of rainfall. The typical situation is that on 16 August, the daily rainfall is 17.9 mm, and the water inflow on that day is 1200 m$^3$/h. On 17 August, the daily rainfall is 7.1 mm, and the water inflow on that day is 2800 m$^3$/h.

The monitoring data of tunnel water inflow and rainfall are substituted into the correlation coefficient calculation formula (Equation (1)) for calculation:

$$\gamma_{xy} = \frac{\Sigma(x - \bar{x})(y - \bar{y})}{\sqrt{\Sigma(x - \bar{x})^2 \Sigma(y - \bar{y})^2}} \tag{1}$$

where $x$, $y$ represents the tunnel water inflow and rainfall to be calculated, and $\bar{x}$, $\bar{y}$ represents the average value of corresponding variables. $\gamma_{xy}$ represents the correlation coefficient between the two. The correlation coefficient between water inflow and rainfall is 0.29. In the correlation analysis of rainfall and water inrush, since the relationship between water inrush and rainfall in the karst tunnel is affected by many factors, from the data point of view, the appearance of water inrush has an obvious lag compared with rainfall, and the correlation coefficient of 0.29 can fully demonstrate the obvious influence between the two [30].

Analysis of Tunnel Water Inrush Characteristics

(1) The water inflow of the tunnel is large, and the water pressure borne by the lining is large. The estimated maximum amount of initial water inrush is about 4666 m$^3$/h. After the construction of the drainage tunnel, there are still some problems, such as insufficient

drainage capacity, water seepage in the construction joints of the invert, and cracking of the secondary lining of the tunnel. There is karst development in the surrounding rock of the tunnel area, and calcium carbonate crystals appear in the drainage pipe and the cracking of the tunnel's secondary lining.

(2) Water inrush in the tunnel occurs in the rainy season and is strongly associated with rainfall. Monitoring records show that water inundation occurs only after rainfall. After rainfall, water gushing in the tunnel will occur as soon as 8 h, and continuous rainfall will lead to more large-scale water gushing than single-day rainfall.

### 3.1.2. Analysis of Tunnel Water Inrush Mechanism

Through the above analysis, it is believed that the main reason for water inrush is the development of the stratum cracks and karst through which the tunnel passes. When it rains, the rainwater penetrates the cracks and karst of the mountain through the surface. When the tunnel is not excavated, the water entering the mountain is discharged to the surface from south to north through karst pipes and cracks or discharged from the spring at the foot of the mountain into the river. After tunnel excavation, the constructed tunnel passes through the karst water-rich area of the fissure. As shown in Figure 10, after the rainy season, due to the replenishment of karst fissure water by rainfall, a large volume of karst fissure water flows downward through the tunnel position, and the tunnel lining bears a large water pressure. During continuous rainfall, the first few days of rainfall will lead to the mountain having a certain water storage function. If the fissure is filled by groundwater, the mountain tends to be saturated, and its water storage capacity decreases, resulting in the next rainfall when the drainage of groundwater through the tunnel is larger and at a higher water pressure. The lining of the tunnel at this location is unable to withstand the huge water pressure, which is released from the invert, drain pipe, and cross passage of the vehicle, resulting in water inrush in the tunnel. Due to the good groundwater connectivity, the water inrush is large.

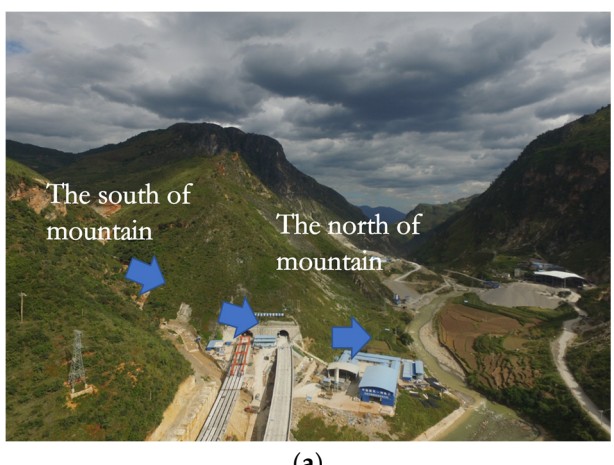
(**a**)

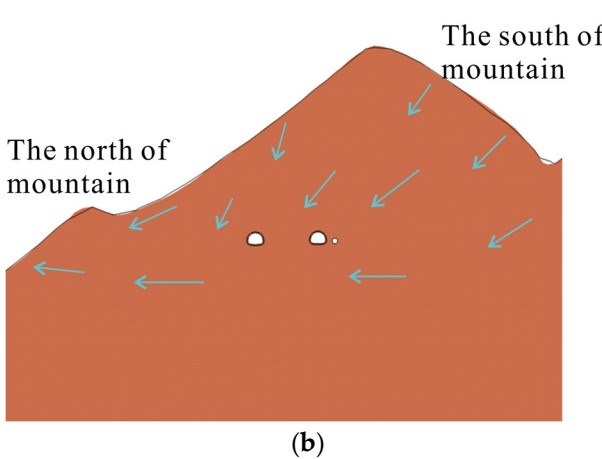
(**b**)

**Figure 10.** Schematic diagram of water inrush mechanism: (**a**) the actual terrain; (**b**) 2D model section.

### 3.2. Study on Prevention and Control Measures of Water Inrush in Tunnel

#### 3.2.1. Determination of Prevention and Control Measures

Because of the mechanism of water inrush in the tunnel, based on the large water volume inrush in this section, the water inrush characteristics of the tunnel, the development of karst around the tunnel, the good connectivity, and the hydrogeological conditions, the water inrush management of the tunnel mainly adopts the drainage scheme.

The drainage tunnel in the previous treatment measures is regarded as the main drainage tunnel. The main drainage tunnel passes through the right line of the tunnel at position K44+605, which is orthogonal to the right line of the tunnel. The top of the drainage tunnel is 8 m away from the bottom of the tunnel, and the main drainage tunnel

passes through the right line and extends 30 m to the right side. A drainage branch is constructed at a position 10 m away from the end of the main drainage tunnel. The branch is parallel to the axis of the tunnel, and the drainage branch is 200 m long. The section of the drainage branch is the same as that of the drainage tunnel.

At the same time, the cavern was revealed to be close to the tunnel, and the water gushing in this section was extremely strong. The cavern is separated from the tunnel by concrete, and the cavern is drained by connecting the cavern to the drainage cavern branch, so that the karst water flowing through it can be discharged from the drainage cavern in time and will not have a large impact on the tunnel (Figure 11).

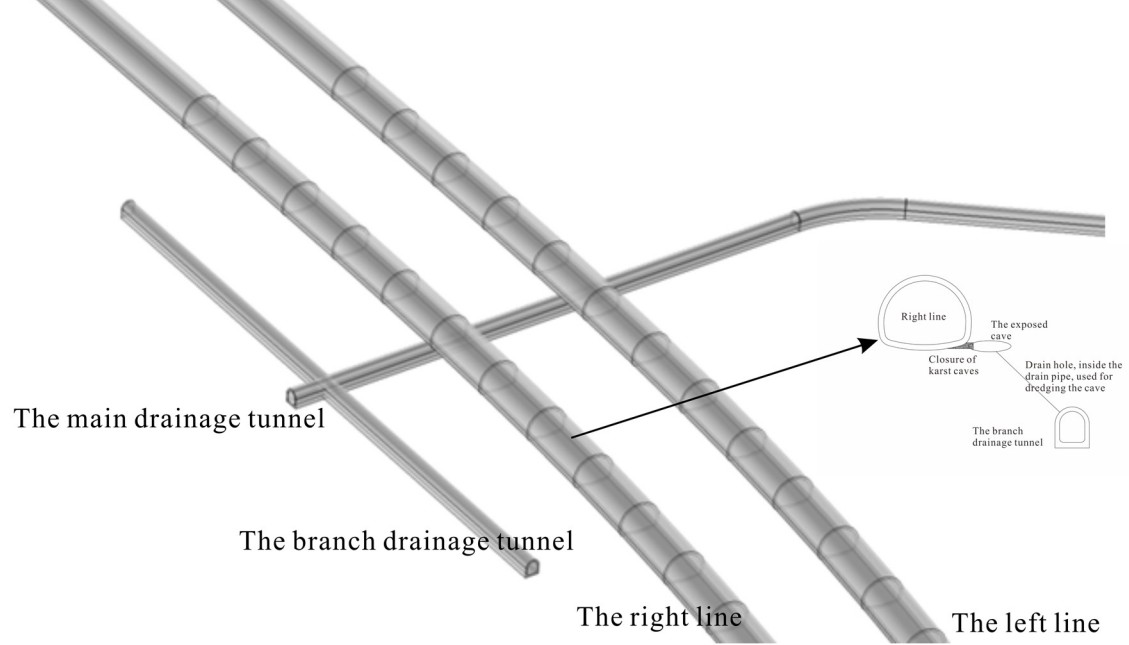

**Figure 11.** Schematic diagram of prevention and control measures for water inrush in tunnel.

### 3.2.2. Numerical Simulation of Prevention and Control Measures
Numerical Simulation Calculation Model of Prevention and Control Measures

For the karst tunnel water-gushing problem, the common numerical simulation software includes COMSOL Multiphysics, Midas, PFC, FLAC$^{3D}$, FLUENT, Visual Modflow, and so on [31–38]. In this paper, COMSOL Multiphysics numerical simulation software was used for calculations. COMSOL Multiphysics has the characteristics of convenient operation of basic functions and common multi-physical field coupling calculations. At the same time, the software has good expansibility and can meet the application of many scenes.

The contour data of the study area were intercepted from the CAD file and imported into the numerical simulation software COMSOL Multiphysics to construct 3D terrain. The tunnel model is constructed according to the construction data in CAD software. Caves in the model according to the result of geological radar to detect the position of decorate, because in addition to the exposed karst cave, the cave is not clear the specific shape and size, so in the center of the detected with a radius of 2 m of the spherical cavity to replace of karst cave, the cave with middle high 12 m in the cylinder, two head of radius 2 m hemispherical approximation instead.

The geometric modeling function of the software is used to add the actual size of tunnels, karst caves, and drainage caves to the actual size of the three-dimensional terrain to build the three-dimensional geological model as shown in Figure 11. The calculation model is 750 m (north-south direction) ×750 m (east-west direction) ×510 m (vertical direction). The model in Figure 12 is divided into 701,131 units in the calculation.

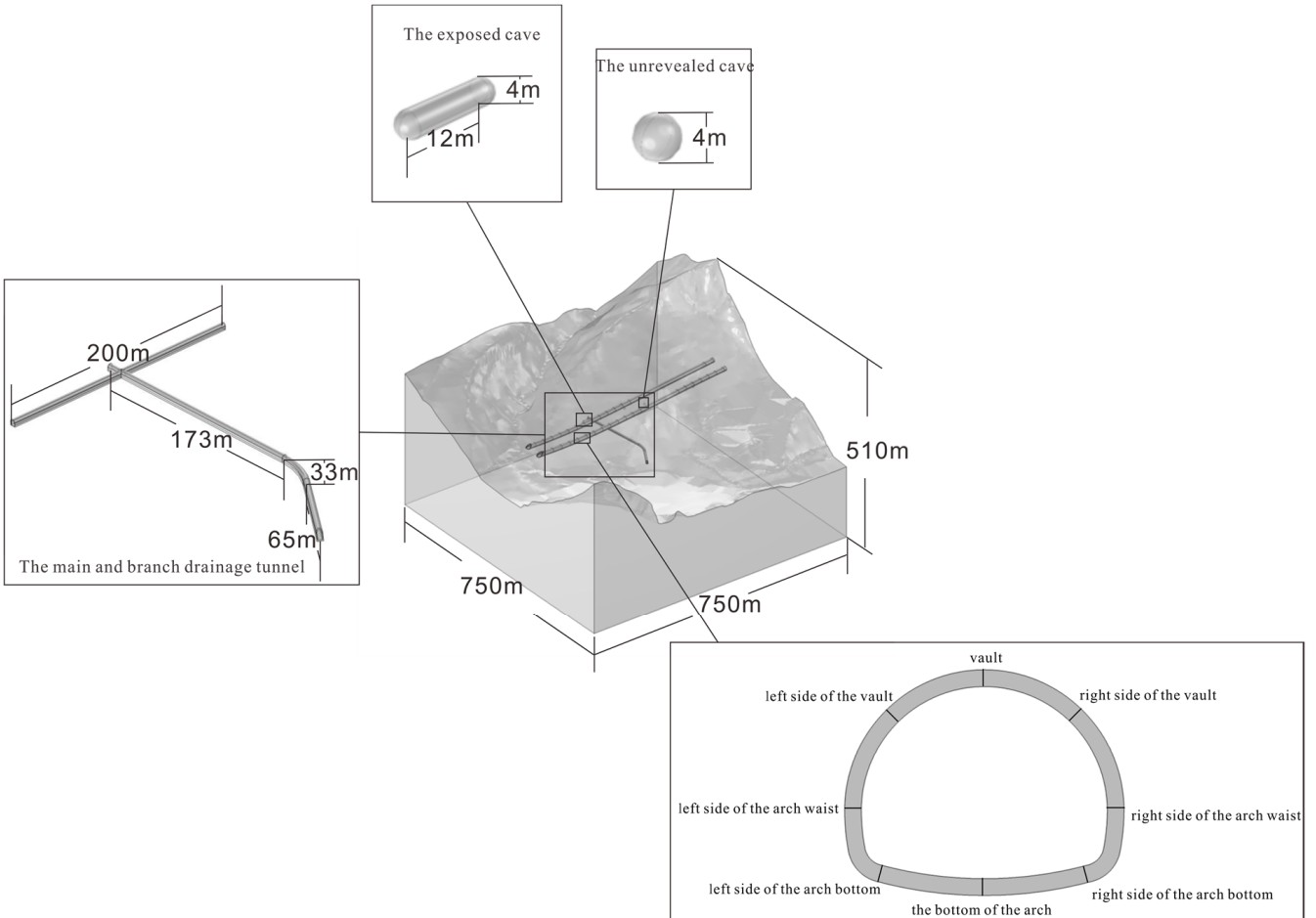

**Figure 12.** 3D geological model.

The linear elastic model is used for the drainage tunnel, the elastoplastic model according to the Mohr–Coulomb strength criterion is used for the rock and soil mass, and the linear elastic model is used for the tunnel lining. The fluid is calculated using Darcy's law.

This calculation needs to evaluate the optimal design of prevention and control measures, and the following three working conditions are intended to be used to analyze the water pressure of the tunnel lining, as well as the internal force and deformation of the tunnel lining.

Working condition 1: The basic working condition under the action of water and karst cave is not considered, and only the 3D geological body and tunnel model are included. Under this working condition, the tunnel lining does not bear water pressure, which is used for comparative analysis with the other two working conditions.

Working condition 2: Under the condition of continuous rainfall, the mountain is saturated, karst is developed in the mountain, and the water inrush in the tunnel has not been treated.

Working condition 3: Consider the working condition after continuous rainfall, mountain saturation, karst development in the mountain, and the implementation of optimized treatment measures.

In terms of solid mechanics, the three-dimensional calculation model of working condition two is supported by rolls around, and only its normal displacement is constrained. The fixed boundary condition is applied to the bottom, the free boundary condition is applied to the upper surface of the model, and the inner surface of the tunnel is the free boundary. The three-dimensional calculation model of working condition three is

supported by rolls around, and only its normal displacement is constrained. A fixed boundary condition (no displacement allowed) is applied to the bottom of the model, and a free boundary condition is applied to the upper surface of the model. The inner surface of the tunnel and drainage tunnel is the free boundary.

In terms of fluid mechanics, no flow is adopted around and at the bottom of the 3D calculation model for working condition 2, and the boundary conditions on the upper surface of the model are set according to the working condition. No flow is adopted around and at the bottom of the 3D calculation model of working condition 3, and the water head at each point on the upper surface is the same as its elevation. The outer boundary of the drainage tunnel is set as a 0 water pressure boundary due to the better drainage measures adopted. The water pressure on the inner surface of the cave and the water pressure of the tunnel lining are set according to the different site conditions [39–41].

The exposed karst cave will be dredged after treatment measures, and the water pressure will be set to 0 water pressure boundary. In the other five karst caves, due to the treatment of the nearby tunnel lining, the drainage measures were strengthened, and the water pressure was simulated without a karst cave.

In the process of numerical simulation, in order to obtain reasonable calculation parameters, the rock blocks in the strata traversed by the tunnel were sampled on site. The rock uniaxial compression test, rock triaxial compression test, and Brazil splitting test were conducted to determine the relevant strength parameters of the rock blocks.

The experimental results were sorted out, and empirical values were obtained for the density and elastic modulus of the lining material and the permeability coefficient of dolomite and the lining [42–44]. The specific calculation parameters are shown in Table 1.

**Table 1.** Mechanical parameters and permeability coefficient.

| Material | Density (kg/m$^3$) | Elasticity Modulus (GPa) | Cohesive Force (MPa) | Internal Friction Angle (°) | Tensile Strength (MPa) | Permeability Coefficient (m/s) |
|---|---|---|---|---|---|---|
| Dolomite | 2702 | 15.4 | 12.62 | 51.2 | 6.23 | $3.75 \times 10^{-5}$ |
| Lining | 2500 | 30 | — | — | — | $2 \times 10^{-11}$ |

Analysis of Numerical Simulation Results of Control Measures

(1)    Variation of water pressure distribution in tunnel and surrounding rock

The karst caves located 50 m, 105 m, 180 m, 295 m, 343 m, and 492 m away from the tunnel exit, surrounding rocks, and tunnel lining are analyzed. From the 3D calculation results of working condition 2 (water inrush prevention is not carried out) and working condition 3 (water inrush prevention is carried out), sections that pass through the cave and are perpendicular to the tunnel axis are selected for analysis of the calculation results. Figure 13 shows the positions of each section in the 3D calculation model of working condition 3 (water inrush prevention is carried out).

Figure 14 is the 3D cloud diagram of water pressure in the calculation results of the 3D model, from which it can be seen that the treatment of water-gushing measures has no significant influence on the whole region.

The karst caves 50 m, 105 m, 180 m, and 295 m away from the tunnel exit are closer to the right line of the tunnel, and the water pressure values of the 8 monitoring sites on the right line of the tunnel are taken for analysis. The karst caves 343 m and 492 m away from the tunnel exit are closer to the left line of the tunnel, and the water pressure values of the 8 monitoring sites on the left line of the tunnel are taken for analysis. The specific position of each point on the tunnel lining is shown in Figure 12.

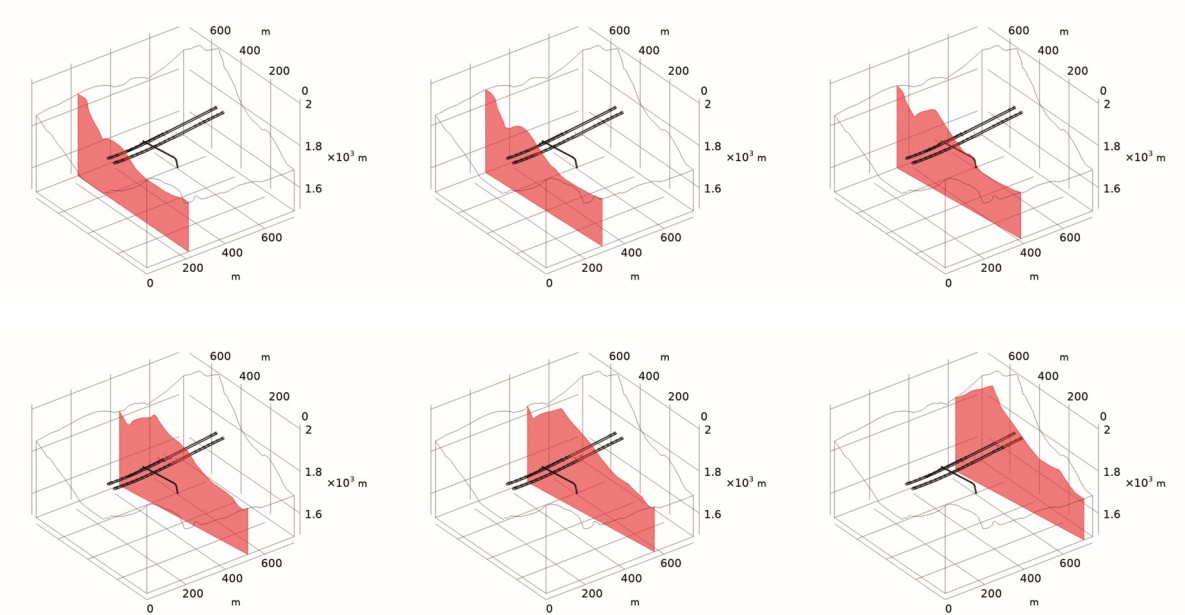

**Figure 13.** Positions of each section in the 3D calculation model of working condition 3.

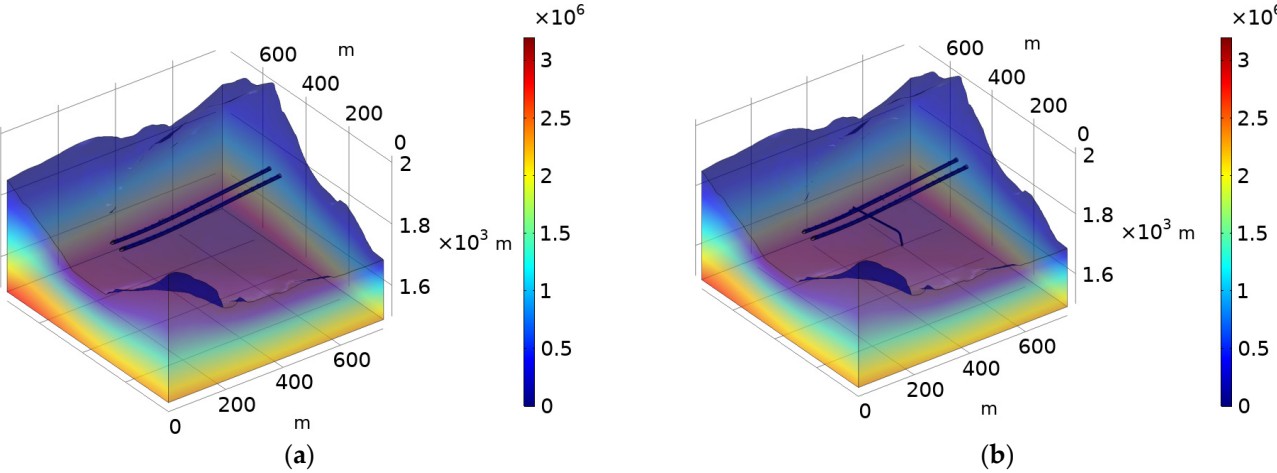

**Figure 14.** Water pressure in the mountain (Unit: Pa): (**a**) no inrush prevention and control was carried out; (**b**) prevention and control of water gushing.

Figure 15 shows the tunnel and surrounding rock water pressure distribution at 50 m from the tunnel exit. Figure 16 shows the water pressure values at each point of the tunnel before and after the treatment. It can be seen that at a distance of 50 m from the tunnel exit, the maximum water pressure of the tunnel before the treatment appears on the right side of the arch bottom, which is 409.6 kPa; after the treatment, the maximum water pressure in the tunnel is still on the right side of the arch bottom, which is 284.1 kPa. The maximum water pressure of the tunnel before and after the treatment decreases by 30.7%. The average water pressure around the tunnel was 320.9 kPa before treatment and 239.3 kPa after treatment. The average water pressure under the tunnel decreased by 25.4% before and after treatment. At the distances of 105 m, 180 m, 295 m, 343 m, and 492 m from the tunnel exit, the maximum water pressure of the tunnel decreases by 39.8%, 84.4%, 39.7%, 25.2%, and 20%, respectively, before and after treatment. The average water pressure of the tunnel decreased by 41.6%, 83.5%, 42.4%, 22.5%, and 14.2%, respectively.

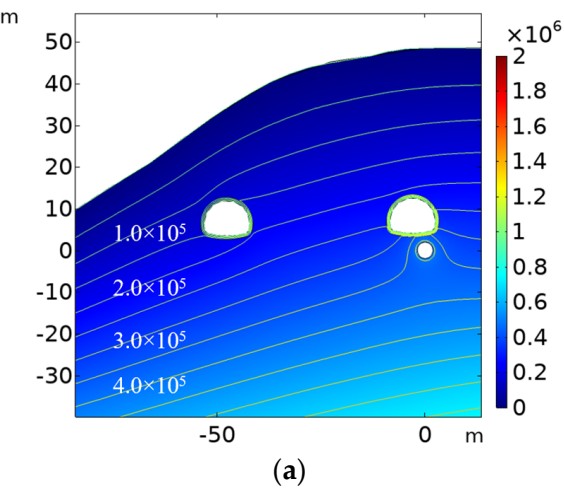
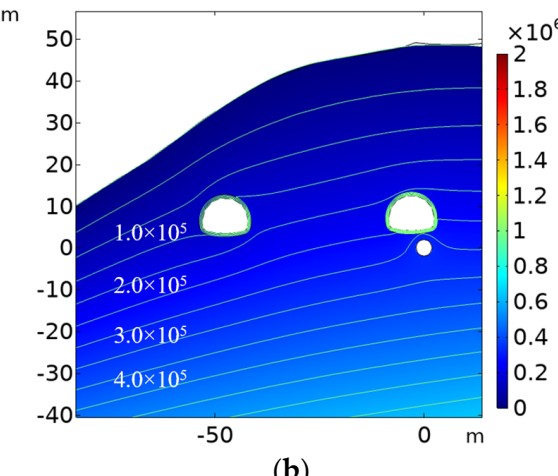

**(a)**                                                        **(b)**

**Figure 15.** Tunnel and surrounding rock water pressure distribution at 50 m of tunnel exit (unit: Pa): (**a**) before treatment; (**b**) after treatment.

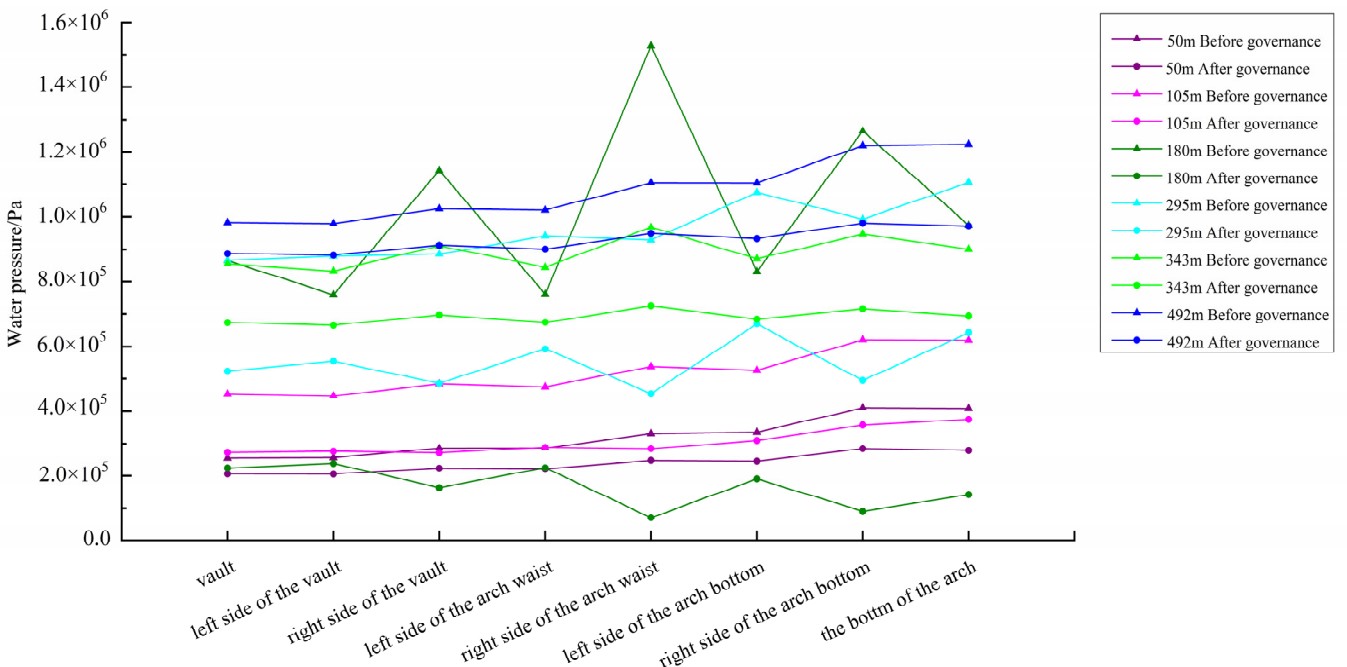

**Figure 16.** Water pressure at each point before and after management.

(2)    Displacement distribution changes of the tunnel and surrounding rock

Figure 17 shows the tunnel and surrounding rock displacement distribution at 50 m of the tunnel exit. Figure 18 shows the specific displacement values of each point on the outside of the tunnel lining. It can be seen that the maximum displacement of the tunnel is 1.04 mm and 1.25 mm, respectively, in the comparison condition (working condition 1) without groundwater and karst cave and the condition without water-gushing treatment (working condition 2) 50 m away from the tunnel exit. After treatment, the maximum displacement of the tunnel is 1.26 mm. The maximum displacement at 105 m, 180 m, 295 m, 343 m, and 492 m away from the tunnel exit is shown in Table 2.

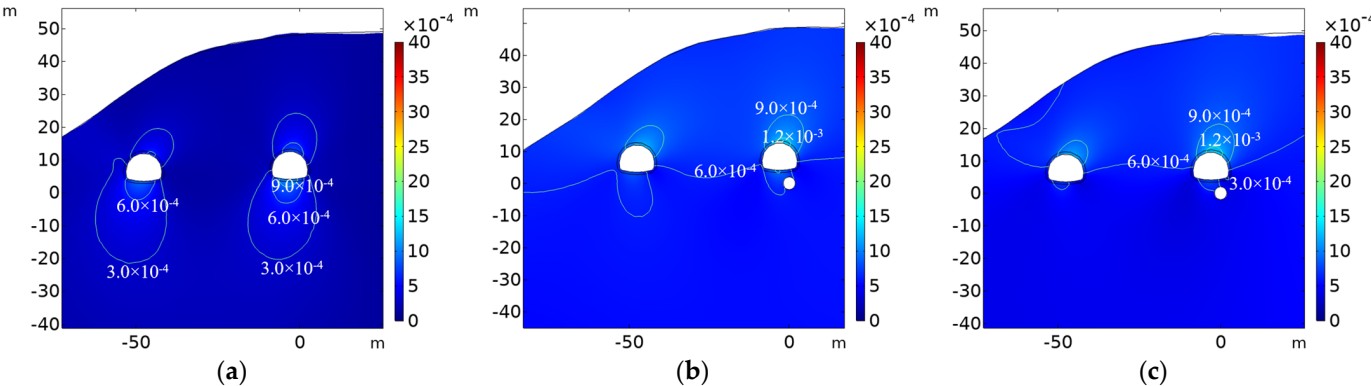

**Figure 17.** Tunnel and surrounding rock displacement distribution at 50 m of tunnel exit (unit: m): (**a**) comparison condition; (**b**) before treatment; (**c**) after treatment.

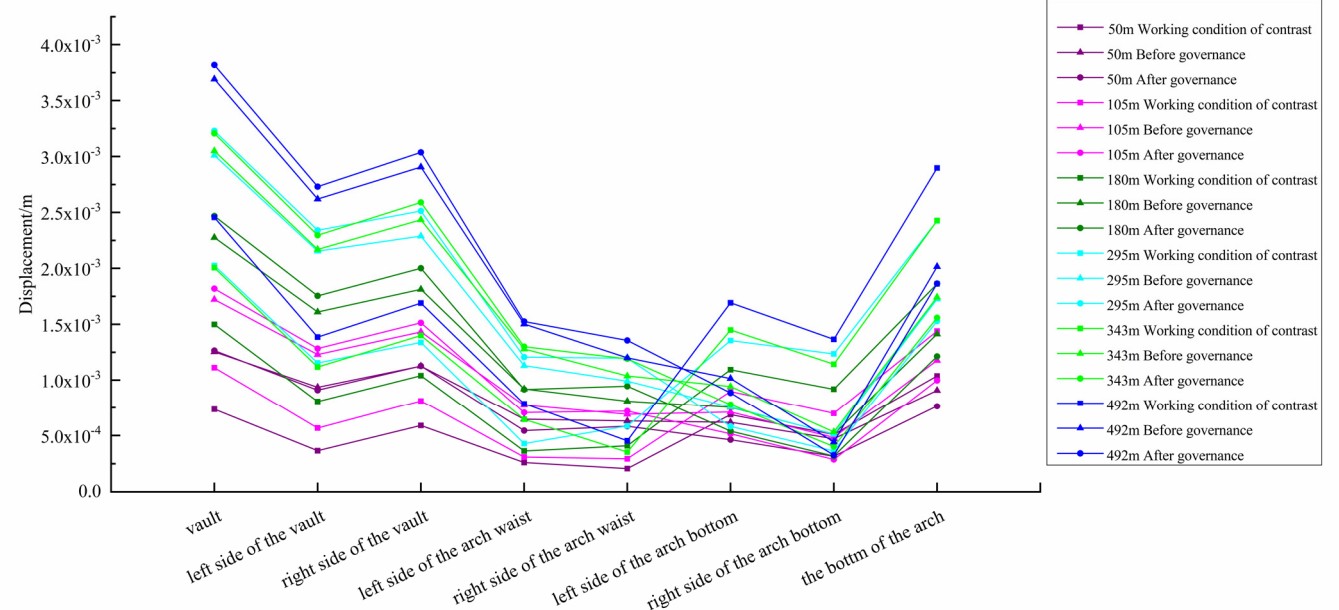

**Figure 18.** Displacement values of each point on the outside of the tunnel lining.

**Table 2.** The maximum displacement of the tunnel.

| Distance from the Tunnel Exit (m) | Working Condition 1 (mm) | Working Condition 2 (mm) | After Treatment (mm) |
|---|---|---|---|
| 50 | 1.04 | 1.25 | 1.26 |
| 105 | 1.44 | 1.72 | 1.82 |
| 180 | 1.86 | 2.28 | 2.47 |
| 295 | 2.43 | 3.01 | 3.22 |
| 343 | 2.43 | 3.05 | 3.21 |
| 492 | 2.90 | 3.69 | 3.82 |

## 4. Discussion

For tunnel water inrush, most of the treatment measures are to grout and plug the water inrush channel, and there are many studies on the water inrush plugging materials and diffusion laws of karst pipelines [45–47]. There is less research on the optimization of established treatment measures to control water inflow in tunnels. Based on the water inrush treatment of the Banyanzi Tunnel of the Huali Highway, this paper proposes the prevention and control of water inrush by using a drainage hole and karst cave dredging

scheme for water inrush control in view of the tunnel water inrush mechanism and verifies the effectiveness of the method through numerical simulation analysis.

The main factors affecting water inrush in the Banyanzi Tunnel of the Huali Highway in Yunnan Province are stratum fracture, karst development, and rainfall. Tunnel water gushing in the rainy season is associated with rainfall; the occurrence of water compared with rainfall has an obvious lag, its location is higher, and the date of occurrence during continuous rainfall is larger the day after the daily rainfall. The conclusion can be made for the tunnel construction schedule and carry out prevention and control of tunnel water gushing in advance measures that provide certain guidance and advice. Based on the field data, the numerical calculation model of the tunnel under different working conditions was constructed using the finite element method. The values of water pressure distribution in the tunnel and surrounding rock at each monitoring site before and after treatment and the values of displacement distribution in the tunnel and surrounding rock at each monitoring site under different working conditions were calculated. It is verified that tunnel water-gushing prevention and control by using a water drainage cave and cavern dredging program can significantly reduce the water pressure on the tunnel and ensure that the tunnel structure displacement change is within the safe range. This study provides a meaningful, comprehensive method for the treatment of tunnel water inrush, and the effectiveness of the method is verified by numerical calculation, which provides a valuable reference for similar projects. This paper selected a more typical eight monitoring sites for the tunnel water treatment before and after the water pressure and displacement analysis in order to grasp the situation more clearly. Similar projects can be selected on the basis of more monitoring sites for monitoring analysis, improved tunnel and surrounding rock water pressure distribution, and displacement changes to grasp the accuracy and verify the effectiveness of the water treatment measures.

## 5. Conclusions

Taking the Banyanzi Tunnel of the Huali Highway in Yunnan as an engineering background, this paper analyzes the water-gushing mechanism of a tunnel in a karst area. Through numerical simulation, it is verified that the water pressure in the tunnel can be significantly reduced by the water-gushing treatment measures and that the deformation of the tunnel structure is kept within a safe range. Specific research conclusions are as follows:

(1) Through the monitoring of rainfall and water inflow in the tunnel, it was found that the occurrence of water inflow has an obvious lag compared with rainfall. By calculating the correlation coefficient, it is found that there is an obvious influence between them.

(2) The karst development around the tunnel was studied by geological radar detection. The connectivity of mountain gully, tunnel water-gushing cavity, and spring points around the mountain was studied by demonstration test. It was proven that the tunnel water-gushing point and spring hole below the mountain were discharge points after the gully surface water entered the mountain.

(3) Combined with the characteristics and hydrogeological conditions of tunnel water gushing, the mechanism of tunnel water gushing is analyzed. The strata fissures and karst through which the tunnel passes are well developed, and the connectivity is excellent. The tunnel was dug through the karst pipes that drain rainwater down the mountain. After rainfall, the rainwater flows through the karst pipeline here, causing large water pressure in the tunnel. The water pressure is released from the invert of the tunnel, the drainage pipe, and the transverse passage of the vehicle, forming the water inrush phenomenon in the tunnel.

(4) In view of the water inrush mechanism, the measures to control the water inrush in the tunnel are optimized. The methods include supporting the drainage tunnel, dredging the karst cave, increasing the drainage capacity, reinforcing part of the section by grouting, and constructing the filling pile under the invert.

(5)   Through numerical simulation analysis, it is verified that the drainage tunnel and karst cave dredging scheme can effectively reduce the water pressure borne by the tunnel; the average water pressure can be reduced by 1.02 MPa to 168.1 kPa (average 83.5%); the deformation of the tunnel structure is kept within the safe range; and the maximum deformation is not more than 4 mm.

**Author Contributions:** This paper was completed under the guidance of H.W., J.Z., H.S. and L.L. established the numerical model; Y.H. analyzed the results and completed the writing of most of the manuscripts; B.Z. contributed to the concept, thought and organization of the study; writing—review; editing, Y.H. All authors have read and agreed to the published version of the manuscript.

**Funding:** This research was funded by the National Natural Science Foundation of China (Nos.42107201 and 41972300), and the Fundamental Research Funds for the Central Universities of China (No. 292019072).

**Institutional Review Board Statement:** Not applicable.

**Informed Consent Statement:** Not applicable.

**Data Availability Statement:** The data used in this study are available on request from the corresponding author.

**Acknowledgments:** The authors would like to thank the reviewers for their constructive comments, which improved the paper.

**Conflicts of Interest:** The authors declare no conflict of interest.

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
