# Peer review of "Water Inrush Mechanism and Treatment Measures in Huali Highway Banyanzi Tunnel—A Case Study"

_water, doi:10.3390/w15030551_

Round 1

Reviewer 1 Report

The authors studied the water inrush characteristics, water inrush mechanism and treatment measures of the karst tunnel based on the project of the Huali Highway Banyanzi Tunnel in China. The paper is the relevance to the scope of journal. Doubtful or controversial arguments were not detected in the paper. The paper has original content and worthy for publication in the journal. I can recommend it for a possible publication. However, following comments must be considered carefully before this recommendation.

1. Abstract needs to be improved. Background is too long. English should be polished. “Based on the analysis of existing data, through field investigation, data monitoring, geological radar detection, tracer test, and the method of numerical simulation, Analyze the characteristics and hydrogeological conditions of tunnel water surges, further summarize the mechanism of tunnel water surges. For the tunnel water gushing mechanism, optimize the tunnel water gushing management measures, and verification.”

2. The conclusions are rather a summary.

3. Figures 1~3 are missing in the text.

4. Line 108, 3.1.1. Engineering geological conditions. It is too verbose.

5. Figures are not clear enough, and they should be improved.

Reviewer 2 Report

Generally, based on a case study, the manuscript discussed the water inrush mechanism in a highway tunnel located in the middle of Lijiang City, China. Also, through numerical simulation using COMSOL software, the influence of the drainage tunnel on the performance of the main tunnel was investigated. The manuscript has a clear logical structure and can be published after some corrections.

1- Introduction. Besides the traffic tunnels, the water transmission tunnels are also facing many challenges such as lining and surrounding rock cracking due to the water-rock and water-lining interactions and pore pressures. Adding some previous numerical research related to those engineering challenges will enrich this section further. For example:

- Investigating the effects of transient flow in concrete-lined pressure tunnels and developing a new analytical formula for pressure wave velocity, published in Tunnelling and Underground Space Technology 2019.

- Mechanical-Hydraulic Interaction in the Cracking Process of Pressure Tunnel Linings, published in International Journal on Hydropower and Dams 2013.

- Stress intensity factors for axial semi-elliptical surface cracks and embedded elliptical cracks at longitudinal butt welded joints of steel-lined pressure tunnels and shafts considering weld shape, published in Engineering Fracture Mechanics, 2017.

- Effect of overburden height on hydraulic fracturing of concrete-lined pressure tunnels excavated in intact rock: a numerical study, published in fluids 2019.

2- 3.3.2 – Was the tunnel excavation phase considered in the numerical modeling?

3- Table 1- The internal friction angle of dolomite looks too high. Are these numbers based on laboratory tests on the surrounding rock in the project? Or previous studies?

4- Figure 17- Since the plastic behavior was assigned to the rock (Table 1), please clarify why the authors used displacement values for investigations, whereas the plastic strains for rock as brittle material should be considered.

5- In the conclusion section, it may be better to separate each identified conclusion with bullets or a list. This way, the reader will have a clearer visual appreciation of the individual conclusion.

Reviewer 3 Report

This study presents a comprehensive research on water Inrush mechanism and treatment measures for Huali Highway Banyanzi Tunnel.

Based on the overall quality, a major revision is needed.

Comments:

(1) In the introduction section, the research gap about the water inrush prevention measures for tunnels should be clarified.

(2) What is your objective of this study. Please highlight and clarify your aim in the introduction.

(3) More recent tunnel cases studies should be included in the introduction. For example, the study "Study on Risks and Countermeasures of Shallow Biogas during Construction of Metro Tunnels by Shield Boring Machine" presented the measures to prevent water inrush for underwater tunnel. In addition, water inrush for special shaped tunnel is of great importance. The authors can also discuss some cases in the introduction. "Case Study of the Largest Concrete Earth Pressure Balance Pipe-Jacking Project in the World". The water inrush prevention for pipe-jacking tunnel was introduced in this study. Therefore, more cases should be discussed in the introduction to show the importance of your study.

(4) For the water inrush analysis for tunnels, did you find or conclude any calculation equations? These could provide a reference for future application.

(5) Figure 9. How did you make sure that the monitoring data was accurate?

(6) Figure 18. The lines are hard to tell. Could you re-draw this figure?

(7) Could you show the limitations of your numerical simulation and your future research plan?

(8) Conclusions. The conclusions should highlight your findings point by point, making your conclusion section clear.

(9) Please polish the language.

Round 2

Reviewer 3 Report

Thanks for your improvement!

Author Response

Thanks for reviewer's suggestions!